# RETRACTED: Involvement of GPR43 Receptor in Effect of *Lacticaseibacillus rhamnosus* on Murine Steroid Resistant Chronic Obstructive Pulmonary Disease: Relevance to Pro-Inflammatory Mediators and Oxidative Stress in Human Macrophages

**DOI:** 10.3390/nu16101509

**Published:** 2024-05-16

**Authors:** Ana Karolina Sá, Fabiana Olímpio, Jessica Vasconcelos, Paloma Rosa, Hugo Caire Faria Neto, Carlos Rocha, Maurício Frota Camacho, Uilla Barcick, Andre Zelanis, Flavio Aimbire

**Affiliations:** 1Department of Medicine, Postgraduate Program in Translational Medicine, Federal University of São Paulo (UNIFESP), Rua Pedro De Toledo 720–2 Andar, Vila Clementino, São Paulo 04039-002, Brazil; ana.karolina25@unifesp.br (A.K.S.); fabiana.olimpio@unifesp.br (F.O.); jessica.carvalho07@unifesp.br (J.V.); cristina.paloma@unifesp.br (P.R.); 2Laboratory of Immunopharmacology, Institute of Science and Technology, Federal University of São Paulo, Rua Talim, 330, Vila Nair, São José dos Campos 12231-280, Brazil; 3Laboratory of Immunopharmacology, Oswaldo Cruz Foundation Fundação Oswaldo Cruz, Av. Brazil, Rio de Janeiro 4036, Brazil; hugo.caire@fiocruz.br; 4Medical School, Group of Phytocomplexes and Cell Signaling, Anhembi Morumbi University, São José dos Campos 04039-002, Brazil; carlosrocha.hd@gmail.com; 5Functional Proteomics Laboratory, Institute of Science and Technology, Federal University of São Paulo, São José dos Campos 12231-280, Brazil; mauriciof.camacho@gmail.com (M.F.C.); lila.barcick@gmail.com (U.B.); andre.zelanis@unifesp.br (A.Z.); 6Postgraduate Program in Pharmaceutical Sciences, Evangelical University of Goiás (UniEvangélica), Avenida Universitária Km 3,5, Anápolis 75083-515, Brazil

**Keywords:** COPD, steroid resistance, lung inflammation, oxidative stress, macrophages, probiotic, GPR43 receptor

## Abstract

Background: Cytokine storm and oxidative stress are present in chronic obstructive pulmonary disease (COPD). Individuals with COPD present high levels of NF-κB-associated cytokines and pro-oxidant agents as well as low levels of Nrf2-associated antioxidants. This condition creates a steroid-resistant inflammatory microenvironment. *Lacticaseibacillus rhamnosus* (Lr) is a known anti-cytokine in lung diseases; however, the effect of Lr on lung inflammation and oxidative stress in steroid-resistant COPD mice remains unknown. Objective: Thus, we investigated the Lr effect on lung inflammation and oxidative stress in mice and macrophages exposed to cigarette smoke extract (CSE) and unresponsive to steroids. Methods: Mice and macrophages received dexamethasone or GLPG-094 (a GPR43 inhibitor), and only the macrophages received butyrate (but), all treatments being given before CSE. Lung inflammation was evaluated from the leukocyte population, airway remodeling, cytokines, and NF-κB. Oxidative stress disturbance was measured from ROS, 8-isoprostane, NADPH oxidase, TBARS, SOD, catalase, HO-1, and Nrf2. Results: Lr attenuated cellularity, mucus, collagen, cytokines, ROS, 8-isoprostane, NADPH oxidase, and TBARS. Otherwise, SOD, catalase, HO-1, and Nrf2 were upregulated in Lr-treated COPD mice. Anti-cytokine and antioxidant effects of butyrate also occurred in CSE-exposed macrophages. GLPG-094 rendered Lr and butyrate less effective. Conclusions: Lr attenuates lung inflammation and oxidative stress in COPD mice, suggesting the presence of a GPR43 receptor-dependent mechanism also found in macrophages.

## 1. Introduction

Chronic obstructive pulmonary disease (COPD) is distinguished by gradual airway inflammation accompanied by a significant influx of inflammatory cells towards the lungs, remodeling of the airways, compromised respiratory mechanics, and decreased airflow to the alveoli [1,2,3].

A quite relevant fact regarding smoking is that, even with high doses of corticosteroids, the clinical symptoms and inflammatory response in patients with COPD remain largely resistant to their anti-inflammatory effects [4,5,6]. Evidence indicates that this condition involves an epigenetic change in histone deacetylase (HDAC), which results in a drastic reduction in HDAC2 activity with subsequent dysregulation of steroid signaling [7,8]. In fact, some authors identify a super-acetylation of the steroid receptor, due to a fall in HDAC activity, as the protagonist of a steroid-resistant condition in COPD [9,10], making the cigarette smoke-induced lung inflammation model a suitable tool to investigate corticosteroid resistance [11,12].

Despite the involvement of neutrophils in resistance to corticosteroid treatment in individuals with COPD, innate immunity cells such as macrophages play a pivotal role in both the initial and later phases of the inflammatory response in COPD [13]. Their activation is intricately controlled by the immune response to pathogens or tissue damage [14,15]. Consequently, when exposed to harmful particles, macrophages can become activated, leading to the secretion of elevated levels of pro-inflammatory cytokines. This, in turn, intensifies the migration of inflammatory cells toward lung tissue [16,17].

The significant contribution of cells involved in the innate immune response in orchestrating inflammation within COPD is widely recognized. These cells release pro-inflammatory mediators such as MCP-1, MIP-1, and RANTES and oxidative stress-related molecules, thus playing a pivotal role [18,19]. Among the primary signaling pathways accountable for the persistent and advancing lung inflammation observed in COPD, the escalation of lung inflammation as well as the oxidative stress stands out, particularly within macrophages [20]. Consequently, macrophages assume a critical responsibility in both the pro-inflammatory response and the unbalance between pro- and anti-oxidant responses in the context of COPD [21,22].

In the context of oxidative stress metabolism-related cell signaling, some authors have demonstrated that exposure to cigarette smoke can diminish the activity of antioxidant enzymes, which are essential for counteracting the surge in oxidant activity associated with COPD [23,24]. Notably, heightened levels of reactive oxygen species (ROS), 3-nitrotyrosine, and 8-isoprostane have been identified within macrophages of individuals with COPD [25,26]. Among the pivotal antioxidant enzymes, catalase (CAT), superoxide dismutase (SOD), glutathione peroxidase (GPx), and glutathione reductase (Gr) hold prominence [27]. These enzymes are under the control of erythroid-related nuclear factor 2 (Nrf2), an antioxidant defense regulatory transcription factor that governs the transcriptional regulation of antioxidant genes by binding to antioxidant response elements [28]. The same authors further elucidate that Nrf2 is intricately connected with the regulation and release of SOD and catalase and heme oxygenase-1 (HO-1) [29]. HO-1 functions as a safeguarding enzyme, countering oxidative stress, and its expression experiences a decline in COPD cases [30]. Hence, the modulation of antioxidants emerges as a crucial factor in the pathophysiology of COPD, working to offset the aggravation of oxidative stress engendered by exposure to cigarette smoke.

A significant interplay exists between the NF-κB transcription factor and oxidative stress metabolism in individuals with COPD. This interplay emerges from the fact that the secretion of ROS triggers the activation of NF-κB, subsequently leading to the release of pro-inflammatory cytokines such as TNF-α, IL-1β, IL-6, GM-CSF, and TSLP [31,32,33]. These cytokines play a contributory role in sustaining the persistent pro-inflammatory response characteristic of COPD, as they function as attractants for neutrophils that secrete proteases [34]. Furthermore, the pro-inflammatory environment within the lungs leads to leukocytes’ resistance to the anti-inflammatory effects of steroids [35]. The cytokines secreted from macrophages during oxidative stress in COPD also prompt the activation of metalloproteases, which degrade the extracellular matrix of the lung. Consequently, a strong correlation is observed between individuals with COPD and extensive airway remodeling [36]. In light of these insights, macrophages are a focal point in endeavors to temper the pro-inflammatory response and mitigate the exacerbation of oxidative stress in individuals with COPD.

Due to the existing limitations in COPD treatment, there is a growing necessity for innovative approaches [37]. In this regard, numerous studies have highlighted the favorable outcomes associated with probiotics—beneficial bacterial strains—in mitigating symptoms linked to both asthma and COPD [38,39]. These probiotics have demonstrated effectiveness in improving the condition of individuals with inflammatory bowel syndrome, thereby reducing clinical manifestations [40]. Considering the concept of the gut–lung axis, it is reasonable to suggest that changes in the gut microbiota could influence the immune response in COPD. Disruptions in the gut microbiota may not only compromise the immune response in the gut but also affect distant organs, including the lungs [41,42]. Despite the acknowledged importance of probiotics in respiratory conditions, there is a scarcity of studies examining changes in the gut microbiota and the mechanisms through which Lacticaseibacillus rhamnosus affects cytokine storms and oxidative stress in the context of COPD.

Hence, the objective of this study was to explore the impact of orally administered Lr on airway inflammation and oxidative stress in a murine model of steroid-resistant COPD. Through this investigation, we sought to uncover potential mechanisms that contribute to these effects.

## 2. Material and Methods

### 2.1. Animals

C57Bl/6 male mice, aged 6–8 weeks (7 mice per group), were procured from the Development Center of Experimental Models for Medicine and Biology at the Federal University of São Paulo. They were housed under specific pathogen-free conditions, with standard environmental parameters including temperature (22–25 °C), relative humidity (40–60%), and a 12 h light/dark cycle, and provided with ad libitum access to food and water. The experimental protocol was approved by the Ethical Committee on Animal Research at the Federal University of São Paulo (protocol number 2166240117).

### 2.2. Induction of COPD

COPD was induced in C57Bl/6 mice by exposing them to cigarette smoke inhalation for 60 days. They were exposed to 14 cigarettes per day, twice daily, for 30 min each session, as depicted in Figure 1A. The cigarette smoke was generated by burning 14 commercial Marlboro cigarettes (tar: 13 mg; nicotine: 1.10 mg; carbon monoxide: 10 mg) obtained from British-American Tobacco Plc. The smoke was directed into a plastic chamber measuring 42 cm (length) × 28 cm (width) × 27 cm (height), where the mice were housed and passively exposed to the cigarette smoke.

### 2.3. Oral Feeding with Lacticaseibacillus rhamnosus

The mice received treatment with probiotic Lacticaseibacillus rhamnosus (Lr) at a dose of 10^9^ CFU per mouse via gavage (Vitalis Laboratory, São Paulo, SP, Brazil) once daily for one week prior to COPD induction. Following COPD induction, the mice continued to receive Lr treatment at the same dose (10^9^ CFU per mouse via gavage) seven days a week until euthanasia, as illustrated in Figure 1A.

### 2.4. Isolation of Peripheral Blood Mononuclear Cells and Macrophage Differentiation

Peripheral blood mononuclear cells (PBMCs) were purchased from Lonza (Catalog #: CC-2702) and were isolated by apheresis and density gradient separation, with ≥ 50 million viable cells per ampoule. For the isolation of monocytes from the PBMCs, we used the magnetic immunosorting method with beads (STEMCELL Technologies, Cambridge, United Kingdom) coated with antibodies against the CD14 surface protein. Briefly, PBMCs were recovered in 5 mL of complete medium, containing 120 μL of “beads”, and incubated on ice for 20 min. After incubation, the volume was topped up to 35 mL with the complete medium and the tubes were centrifuged at 300× *g* for 10 min at 4 °C. The supernatant was discarded, and the cells were recovered in 5 mL of complete medium. Then, the cell suspension was added to the Separation Column (STEMCELL Technologies, Cambridge, UK) already properly connected to the magnetic support and with balanced electrical charges, waiting for the elution by gravity. After the first elution, 5 mL of complete medium was added to the column, allowing elution by gravity again. To obtain monocytes, or CD14+ cells, the separation column was removed from the magnetic support and the cell suspension was eluted with 5 mL of complete medium. In the cell suspensions, a volume of complete medium sufficient for 35 mL was added and the tubes were centrifuged at 300× *g* for 10 min at 4 °C. The cells were recovered with complete RPMI medium without FBS, in different volumes, respecting the concentration of 5.0 × 10^6^ cells/mL. Subsequently, the cells were labeled with anti-CD14-FITC (STEMCELL Technologies, Cambridge, United Kingdom) and subjected to analysis by flow cytometry to assess cell purity. For macrophage differentiation, PBMCs (CD14) were incubated with PMA (12 ng/mL) and remained in the oven for 48 h before the experiment. Differentiation was verified through cellular morphology with the aid of an inverted microscope before the experimental test.

### 2.5. Macrophages Expose to Cigarette Smoke Extract and Treated with Butyrate

Macrophages were subjected to 25 µM of butyrate for a period of 2 h. Subsequently, the cells were exposed to 2.5% CSE for 4 h, which constituted the butyrate + CSE group. The CSE was produced by burning a single unfiltered cigarette in 10 mL of culture medium. A vacuum pump operating at a pressure of −11 Kpa was employed to aid in the infusion of cigarette smoke into the culture medium. This experimental configuration is illustrated in Figure 1B.

### 2.6. Treatment with GLPG-094, a Butyrate GPR43 Receptor Inhibitor

In order to investigate whether the butyrate GPR43 receptor is involved in the beneficial effect of Lr, mice were treated with GLPG-094, a butyrate GPR43 receptor inhibitor, in vivo with a dose of 1 mg/Kg 1 h before exposure to Lr. To investigate whether the GPR43 receptor influences the in vitro effect of butyrate, human macrophages were treated with a concentration of 0.1 μM of GPLG-094, 1 h before exposure to butyrate.

### 2.7. Treatment with Dexamethasone

In order to set steroid resistance in vivo and in vitro, mice were treated with dexamethasone at a dose of 10 mg/Kg and human macrophages were treated with dexamethasone at a concentration of 1 μM, both 1 h after exposure to CSE.

### 2.8. Experimental Groups

All mice were housed together in a common enclosure and were randomly assigned to one of four groups, each consisting of seven mice. These groups were as follows: (1) the control group, comprising non-manipulated mice; (2) the COPD group, comprising mice exposed to cigarette smoke; (3) the Lr + COPD group, comprising mice treated with Lr and exposed to CSE; and (4) the GPLG-094 + Lr + CSE group, comprising mice pretreated with GLPG-094 one hour prior to Lr exposure and subsequent exposure to CSE. For in vitro assays, the experimental groups were as follows: (1) the control group, comprising non-manipulated cells; (2) the CSE group, comprising cells exposed to CSE; (3) the butyrate + CSE group, comprising cells treated with butyrate and exposed to CSE; and (4) the GLPG-094 + butyrate + CSE group, comprising macrophages pretreated with 0.1 μM of GLPG-094 one hour before exposure to butyrate, followed by exposure to CSE.

### 2.9. Cell Viability

To assess cell viability, trypan blue staining was conducted. Initially, macrophages were cultured to 90% confluence (1 × 10^6^ cells/well) in 6-well plates using complete RPMI1640 supplemented with 10% fetal bovine serum and 1% penicillin-streptomycin across all experimental groups. Subsequently, cells were exposed to CSE, dexamethasone, butyrate, or GLPG-094, and maintained for 24 h in a humidified incubator at 37 °C with 5% CO_2_. After the incubation period, the cells were washed once with 1 mL of PBS. Then, 500 μL of trypsin solution was added to each well, and the mixture was incubated at 37 °C for approximately 15 min. The cells were then transferred to microfuge tubes using scraping and centrifuged at 5000 rpm for 10 min. The supernatant was discarded, and the cells were resuspended in 1 mL of RPMI 1640, they were thoroughly mixed, and appropriate dilutions were made. Subsequently, equal volumes of resuspended cells and trypan blue solution (0.4% wt/vol) were mixed in a 1:2 ratio. Only cells that excluded trypan blue dye were considered viable. Cell counting was performed using a hemocytometer. To determine the final value of viable cells/mL in the original cell suspension, the count from each quadrant was multiplied by 1 × 10^4^ and then by 2 to correct for the dilution caused by trypan blue addition. The percentage of live cells was calculated by dividing the count of live cells by the total cell count.

### 2.10. Cellularity in BALF

The mice were anesthetized with ketamine (200 mg/kg) and xylazine (10 mg/kg) intraperitoneally and underwent the abdominal aorta bleeding procedure. After euthanizing the mice with excess anesthetics, the trachea was cannulated, and the lungs were rinsed with 0.5 mL of cold PBS (saline). This was followed by two additional washes with the same volume of saline. Total and differential cell counts in the bronchoalveolar lavage fluid (BALF) were determined using a hemocytometer. Differential cell counts in the BALF were determined through cytospin preparation stained with Instant-Prov (Newprov, Brazil). A total of 300 cells were counted to identify macrophages, neutrophils, and lymphocytes using light microscopy.

### 2.11. Histology and Morphometric Analysis

Following euthanasia, the lungs were carefully extracted, perfused, and fixed in 10% paraformaldehyde for histological examination. Lung segments approximately 5 μm thick were stained with hematoxylin and eosin (Sigma-Aldrich Co., St. Louis, MO, USA), Toluidine blue, Verhoeff, Picrosirius, and Periodic Acid-Schiff (PAS). The parameters assessed included mucus secretion and deposition of collagen fibers. Images of five airways from each animal were captured at 400× magnification using an Olympus BX 43 microscope equipped with CellSens Standard software and Image Pro-Plus software (4.5, NIH, Bethesda, MD, USA).

### 2.12. Pro-Inflammatory Mediators

Following euthanasia, bronchoalveolar lavage fluid (BALF) was collected for subsequent ELISA tests. The samples were then centrifuged at 5000 rpm, 4 °C for 15 min, and the supernatant was collected and stored at −80 °C for cytokine measurement. The levels of TNF-α, IL-1β, IL-6, GM-CSF, and TSLP in the BALF were quantified using enzyme-linked immunosorbent assays (ELISA) kits purchased from R&D Systems (Minneapolis, MN, USA). ELISA was performed following the manufacturer’s instructions, and values were expressed as pg/mL. For in vitro assays, supernatant samples from macrophages were processed using a multiplex biometric immunoassay. Monoclonal antibodies specific for the target proteins were used to measure cytokines MCP-1, MIP-1, and RANTES according to the manufacturer’s instructions (Bio-Plex Human Cytokine Assay; Bio-Rad Inc., Hercules, CA, USA).

### 2.13. Measurement of Pro-Oxidant and Antioxidant Agents and Lipid Peroxidation

Lung tissue samples from mice and supernatant samples from human macrophages were analyzed to assess the levels of oxidative damage markers and antioxidant response using enzyme immunoassays. The oxidative response was evaluated by measuring reactive oxygen species (ROS), isoprostane (8-iso-PGF2α), and NADPH oxidase. Lipid peroxidation was determined using the thiobarbituric acid-reactive substances (TBARS) method, with TBARS levels quantified by absorbance at 535 nm and expressed as malondialdehyde (MDA) equivalents (nmol MDA/mg protein). Biomarkers of antioxidant response assessed in lung tissue included superoxide dismutase (SOD), catalase, and heme oxygenase-1 (HO-1). For in vitro assays, supernatant samples from macrophages were processed, and ROS, 8-Iso, TBARS, and SOD were measured using the same methods described for lung tissue samples.

### 2.14. Expression of NF-κB and Nrf2 in Lung

The mRNA expression in lung tissue from mice and human macrophages across all experimental groups was prepared for Real Time-PCR (RT-PCR) analysis of gene expression of NF-κB. The primers used for NF-κB mRNA quantification were forward 5′-CCGGGAGCCTCTAGTGAGAA-3′ and reverse 5′-TCCATTTGTGACCAACTGAACGA-3′. For Nrf2 mRNA quantification, the primers used were forward 5′-TCACACGAGATGAGCTTAGGGCAA-3′ and reverse 5′-TACAGTTCTGGGCGGCGACTTTAT-3′. GAPDH primers were utilized as endogenous controls, with the forward primer sequence being 5′-TTCAACGGCACAGTCAAGG-3′ and the reverse primer sequence being 5′-ACATACTCAGCACCAGCATCAC-3′.

### 2.15. Statistical Analyses

All analyses were performed using GraphPad Prism version 8.0 software. Student’s *t*-test was employed for comparisons between two groups, while the one-way ANOVA test with Tukey’s post-test was utilized for groups with more than two categories. Statistical significance was established at *p* < 0.05.

## 3. Results

### 3.1. Cell Viability

The Table 1 presents the cellular viability of human macrophages exposed to the culture medium or CSE (2.5%) or dexamethasone (1 μM) or butyrate (25 μM), or GLPG-094 (0.1 μM). The cell viability to CSE, dexamethasone, butyrate, and GLPG-094 was not significantly different compared to the control group (culture medium). Therefore, the subtle reduction in cell viability after CSE did not compromise the experimental assays.

### 3.2. Model of COPD Resistant to Corticoid

Patients with chronic obstructive pulmonary disease (COPD) who exhibit predominant neutrophilia and downregulation of histone deacetylase (HDAC2) demonstrate limited responsiveness to pharmacological steroid therapy. In order to confirm the steroid resistance, the COPD mice were treated with dexamethasone, and then cellularity and cytokine secretion in the BALF as well as airway remodeling, were assessed. Table 2 shows that dexamethasone was not effective in reducing the number of inflammatory cells (neutrophils, macrophages and lymphocytes) in the BALF of COPD mice. Similarly, Table 3 shows that dexamethasone-treated COPD mice did not present a change in levels of pro-inflammatory mediators as well as NF-κB expression compared to the COPD mice. In addition, Figure 2 illustrates that COPD mice also presented mucus into bronchus (Figure 2A), airway collagen (Figure 2B), and parenchymal collagen (Figure 2C) in the lungs even after dexamethasone treatment compared to the COPD group (Table 4). In addition, Figure 3 illustrates that HDAC2 activity (3A) in the lungs of COPD mice as well as the protein level (Figure 3B,C) of HDAC2 in macrophages exposed to CSE were drastically reduced.

### 3.3. Probiotics Reduce Lung Inflammation and Airway Remodeling

Leukocyte infiltration, which encompasses the presence of macrophages, neutrophils, and lymphocytes, is a crucial factor in the pathology of chronic obstructive pulmonary disease (COPD). Figure 4 illustrates the impact of Lr on lung inflammation, as assessed by the differential cell counts in bronchoalveolar lavage fluid (BALF). In the COPD group, there was a notable increase in the total cell population (Figure 4A) compared to the control group. Additionally, in the cigarette smoke extract (CSE) group, the numbers of neutrophils (Figure 4B), macrophages (Figure 4C), and lymphocytes (Figure 4D) were significantly elevated compared to the control group. Conversely, administration of Lr led to a significant reduction in the number of infiltrating cells in the lungs of the COPD group, as demonstrated in Figure 4A–D). Given that Lr inhibits the infiltration of neutrophils and macrophages in BALF, we further investigated whether Lr exerted an anti-inflammatory effect on lung tissue. As depicted in Figure 5, histological changes in the lung parenchyma and intrabronchial region were observed in the COPD group and were partially reversed after Lr administration. Specifically, Lr administration attenuated intrabronchial mucus accumulation (Figure 5A) as well as parenchymal (Figure 5B) and peribronchial collagen deposition (Figure 5C) induced by COPD. There was no significant difference observed between the control and Lr groups.

### 3.4. Probiotics Mitigate Pro-Inflammatory Mediators in Lungs

Cytokines are pivotal in chronic obstructive pulmonary disease (COPD) because they stimulate the secretion of pro-inflammatory mediators within the pulmonary environment. Consistent with these findings, our results depicted in Figure 6 demonstrate an increase in the secretion of TNF-α (Figure 6A), IL-1β (Figure 6B), IL-6 (Figure 6C), GM-CSF (Figure 6D), and TSLP (Figure 6E) in bronchoalveolar lavage fluid (BALF), as well as higher levels of NF-κB (Figure 6F) mRNA expression in mice exposed to cigarette smoke extract (CSE) (COPD group) compared to the control. Conversely, Lr treatment attenuated the secretion of pro-inflammatory mediators in BALF and reduced NF-κB expression in the lung tissue of mice in the Lr + COPD group compared to the COPD group. There was no significant difference observed between the control and Lr groups.

### 3.5. Probiotics Modulate Oxidative Stress in Lungs

To explore whether Lr could impact the imbalance between pro- and anti-oxidant agents in COPD mice, we assessed the effect of Lr on secretion of pro-oxidant agents ROS, 8-isoprostane (8-Iso), TBARS, and NADPH oxidase as well as the Lr effect on anti-oxidant agents SOD, catalase, HO-1, and Nrf2 expression. Thus, Figure 7 illustrates that the increased levels of ROS (Figure 7A), 8-Iso (Figure 7B), NADPH oxidase (Figure 7C) and TBARS (Figure 7D) in BALF in lung tissue were higher in mice from the COPD group compared to the control group. Conversely, the Lr-treated COPD mice presented lower levels of these pro-oxidant agents (Figure 7A,B). In a different way, Figure 8 illustrates that COPD mice had a significant reduction in anti-oxidant agents SOD (Figure 8A), catalase (Figure 8B), and HO-1 (Figure 8C) as well as mRNA expression to Nrf2 (Figure 8D) in lung tissue compared to the control group. On the contrary, the levels of these anti-oxidant agents in lung tissue were partially restored in Lr-treated COPD mice in comparison with the COPD group. There is no significant difference between the control and *Lr* groups.

### 3.6. Butyrate Attenuates Pro-Inflammatory Mediator Secretion and Oxidative Stress in Steroid-Resistant Macrophages

In an effort to emulate corticoid resistance to cytokine secretion in macrophages, our findings reveal that dexamethasone did not alter the levels of pro-inflammatory cytokines MCP-1, MIP-1, and RANTES, and nor was NF-κB expression in macrophages challenged with cigarette smoke extract (CSE) (Table 5). Conversely, Figure 9 illustrates that butyrate decreased the secretion of MCP-1 (Figure 9A), MIP-1 (Figure 9B), and RANTES (Figure 9C), as well as NF-κB expression (Figure 9D) in CSE-exposed macrophages. Furthermore, as depicted in Figure 10, butyrate mitigated the secretion of ROS (Figure 10A), 8-Iso (Figure 10B), and TBARS (Figure 10C), while also partially restoring the level of SOD (Figure 10D) and mRNA expression of Nrf2 (Figure 10E) in macrophages exposed to CSE.

### 3.7. GPR43 Receptor Inhibitor Partially Attenuates Effect on In Vivo Lr and In Vitro Butyrate

It is widely recognized that probiotics metabolize dietary fibers to generate short-chain fatty acids, including butyrate. Furthermore, it has been extensively documented that the anti-inflammatory effects of butyrate can be partly impeded by GLPG-094, an inhibitor of the butyrate receptor. Our findings show a significant reduction in the in vivo effect of Lr on the total cell numbers (Figure 11A) and macrophage population (Figure 11B) in BALF, as well as ROS (Figure 11C) concentration and NF-κB (Figure 11D) mRNA expression in lungs from COPD mice treated with GLPG-094 compared with the COPD group. Figure 12 depicts that the butyrate effect on secretion of MCP-1 (Figure 12A), MIP-1 (Figure 12B), RANTES (Figure 12C), ROS (Figure 12D), and NF-κB (Figure 12E) from human macrophages treated with GLPG-094 and exposed to CSE was lower compared with macrophages exposed to both butyrate and CSE. In addition, GLPG-094 also significantly attenuated the restorative effect of butyrate on the HO-1 (Figure 12F) levels as well as expression of Nrf2 (Figure 12G) in CSE-exposed human macrophages.

## 4. Discussion

The novelty that the present study brings refers to the effect of the probiotic *Lacticaseibacillus rhamnosus* on lung inflammation, airway remodeling and oxidative stress in conditions in which animals exposed to cigarette smoke exhibit resistance to pharmacological treatment with the steroid dexamethasone. To test the Lr effect in this condition is important because individuals with COPD indeed exhibit a reduction in the anti-inflammatory response induced by steroids [43]. Our results show that the increase in the population of inflammatory cells, concentration of pro-inflammatory cytokines in the BALF, mucus secretion and collagen deposition were not attenuated by dexamethasone treatment in mice exposed to CSE. In order to confirm the epigenetic feature of steroid resistance in COPD, we found that the HDAC2 activity in the lung tissue of COPD mice was drastically reduced. The involvement of HDAC2 in steroid resistance in different lung diseases is well known [44]. In addition, CSE-exposed human macrophages also presented steroid resistance to pro-inflammatory cytokines and NF-κB expression. Similarly, human macrophages exposed to CSE had a marked decrease in HDAC2 expression. This condition shows that Lr was tested as an anti-inflammatory agent in steroid-resistant COPD.

The progression of COPD is closely associated with cytokine storm as well as a disturbance of oxidative stress, which results in the chronicity of mucus secretion and collagen deposition with subsequent worsening of airway remodeling [45]. Our findings point in this direction, once Lr was able to attenuate the secretion of TNF-α, IL-1β, IL-6, GM-CSF, and TSLP in the BALF of COPD mice. These findings have importance since these cytokines act distinctly in COPD but exhibit synergy in worsening the lung inflammatory condition, as they stimulate neutrophil migration, and increase the Th1 cell population, but mainly activate macrophages, once this condition perpetuates mucus secretion and collagen deposition due to the chronic cycle of injury-repair, which characterizes the exacerbation of airway remodeling [46,47]. In addition, Lr attenuated the unbalance of oxidative stress, reducing the levels of ROS, 8-Iso, and NADPH oxidase as well as lipid peroxidation. The anti-inflammatory and antioxidant effects of some probiotics have been demonstrated in various inflammatory conditions, either in experimental models or even in clinical settings [48,49]. Therefore, some studies show that probiotics are effective in pulmonary inflammation conditions, such as asthma and COPD, since they reduce lung inflammation and airway remodeling [50,51]. However, our findings show that Lr exhibited anti-inflammatory and antioxidant effects in a steroid-resistant condition. These findings deservedly highlight how Lr had a beneficial effect on a condition in which the “gold standard” therapy for chronic diseases produced a limited effect in COPD mice. In addition, Lr significantly attenuated the main cytokines involved in the chronicity of COPD [52].

Some authors have shown that there is a cross-talk between the transcription factor NF-κB, which is known as the master regulator of secretion of pro-inflammatory mediators in diverse chronic diseases, and metabolites of oxidative stress. Our findings show that Lr downregulated the mRNA NF-κB expression and ROS secretion in lungs from COPD mice. In fact, there is evidence that the Lr effect on pro-inflammatory cytokines is due to inhibition of NF-κB [53]. These results help to explain the Lr effect on both the inflammatory cell population and the secretion of TNF-α, IL-1β, IL-6, GM-CSF, and TSLP in the BALF of COPD mice.

Our findings show that Lr exerts effects on different components of cell signaling at a post-transcriptional level, since the expression of NF-κB as well as NADPH oxidase levels were reduced in the lung tissue of COPD mice. These data indicate that the anti-inflammatory effect as well as the anti-oxidant effect of Lr seem to be independent. Whereas corticoid therapy was not effective to attenuate the lung inflammation as well as airway remodeling in COPD mice, and NADPH oxidase is not a molecular target of corticoids, it is reasonable to suggest that Lr presents a wider effect compared to dexamethasone, and maybe for this reason had a beneficial effect in a corticoid-resistance pro-inflammatory condition. Therefore, it is reasonable to suggest that the better effect of Lr on lung inflammation and airway remodeling in a corticoid-resistance COPD condition is also closely linked to the probiotic action in modulating the oxidative stress.

Probiotics can metabolize dietary fiber in the intestine, principally in the colon portion, resulting in the production of short-chain fatty acids, including butyrate. Butyrate is well known for its anti-inflammatory and immunomodulatory effects in various chronic inflammatory diseases, particularly those associated with intestinal inflammatory disorders [54]. This condition has sparked interest in the effect of butyrate on organs distant from the intestine, and indeed several authors have shown a beneficial effect of butyrate in asthma and acute lung inflammation [55,56]. Therefore, we investigated whether butyrate could be involved in the anti-cytokine and anti-remodeling effect of Lr. When butyrate binds to the GPR43 receptor, it triggers a series of cellular responses, including the regulation of immune function, the production of intestinal hormones, and the maintenance of intestinal barrier integrity [57]. Therefore, the GPR43 receptor plays an important role in mediating the beneficial effects of butyrate. Our results support our hypothesis, as mice treated with GPLG-094 (butyrate GPR43 receptor inhibitor) showed a reduction in the anti-inflammatory and antioxidant effect of Lr. In fact, some authors have evidenced that the GPR43 receptor on immune cells is involved in diverse inflammatory conditions, such as the secretion of IL-6 from macrophages exposed to SARS-CoV-2 membrane glycoprotein [58], atherosclerotic inflammation [59], gout [60], pneumoniae infection [61], diabetes [62], adipose tissue M2-type macrophages [63], and acute lung injury [64]. Our findings add to the results of these authors, as in vivo inhibition of the GPR43 receptor decreased the Lr effect on pro-inflammatory mediators TNF-α, IL-1β, IL-6, GM-CSF, and TSLP as well as the transcription factor NF-κB, all of which are involved with the chronicity of COPD. Moreover, there was a significant reduction in Lr ability in restoring the balance of oxidative stress, i.e., Lr decreases ROS, isoprostane, NADPH oxidase, and TBARS, while it slightly increases SOD, catalase, and HO-1, and upregulates Nrf2 in the lungs of COPD mice treated with GLPG-094.

There is a growing body of evidence which supports a major role for macrophages in cytokine storm and disturbances of oxidative stress in COPD, which can lead to steroid resistance [65]. In vivo, our results show a significant reduction in activity and protein expression of HDAC2 in macrophages exposed to CSE, confirming that the mechanism of steroid resistance is present in these cells. In addition, dexamethasone did not change the increase in secretion of pro-inflammatory mediators, mirroring what was observed in the in vivo model. With this thought, we decided to study the response of human macrophages to treatment with butyrate in a condition in which these cells exposed to CSE are resistant to treatment with dexamethasone. Our findings show that the exposure of macrophages to CSE resulted in the increased secretion of MCP-1, MIP-1, and RANTES as well as upregulation of mRNA expression to NF-κB. On the contrary, butyrate attenuated the secretion of ROS, isoprostane, and TBARS in macrophages submitted to CSE. Moreover, Lr restored the levels of SOD as well as mRNA expression for Nrf2. These results were confirmed when human macrophages exposed to CSE and treated with GLPG-094 presented a reduction in the butyrate effect on both the secretion of pro-inflammatory mediators and the unbalance of oxidative stress. Similarly to in vivo findings with Lr, the in vitro effect of butyrate was also less effective on transcription factors NF-κB and Nrf2 when human macrophages were pretreated with GLPG-094.

Currently, some authors demonstrate that butyrate interferes with cellular signaling involving the pro-inflammatory transcription factor NF-κB and the antioxidant transcription factor Nrf2 as a mechanism of action to attenuate allergic asthma [66] and the secretion of pro-inflammatory mediators from epithelial cells after exposure to LPS [67]. These authors identified that the butyrate receptor GPR43 is partially responsible for the anti-inflammatory effects of this SCFA, as the anti-inflammatory effect of butyrate may also be associated with its effect on histone acetylation via the inhibition of histone deacetylases [68]. Our results corroborate these findings since in the presence of the GPR43 receptor inhibitor, GLPG-094, the butyrate effect on cytokine secretion and unbalance of oxidative stress was not completely blocked. Therefore, the crosstalk of these signaling pathways with transcriptional factors NF-κB and Nrf2 appears to be an important mechanism of the protective effect of butyrate on inflammatory diseases.

Despite the present study not assessing other potential cellular signaling pathways that result from the anti-inflammatory and antioxidant effects of Lr and butyrate, it is important to highlight that the present manuscript demonstrates for the first time the involvement of the GPR43 receptor in the anti-inflammatory and antioxidant effect of Lr in a condition where mice with COPD and human macrophages exposed to CSE are resistant to steroids.

## 5. Conclusions

Finally, the beneficial effect of Lr in suppressing the secretion of pro-inflammatory mediators and modulating oxidative stress establishes a promising environment for the use of probiotics in individuals with COPD, especially considering the crucial role that macrophages play in orchestrating the chronicity and progression of steroid-resistant COPD.

## Figures and Tables

**Figure 1 nutrients-16-01509-f001:** (**A**) in vivo: COPD was induced in C57Bl/6 mice through exposure to cigarette smoke inhalation for 60 days, twice daily, with each session lasting 30 min. This regimen was administered 7 days a week, totaling 14 cigarettes per day. Commercial Marlboro cigarettes were used to generate the smoke. The smoke was introduced into a plastic box where the mice were housed, allowing them to passively inhale the cigarette smoke. Prior to COPD induction, the mice were administered 1 × 10^9^ CFU of probiotic Lacticaseibacillus rhamnosus (Lr) once daily for one week. Following COPD induction, the mice continued to receive Lr treatment 7 days a week until euthanasia. (**B**) in vitro: macrophages were incubated with 25 µM of butyrate for 2 h. Two hours after butyrate addition, the cells were stimulated with 2.5% of CSE during 4 h.

**Figure 2 nutrients-16-01509-f002:** Photomicrographs of lung tissue. (**A**) intrabronchial mucus, (**B**) peribronchial collagen, and (**C**) parenchymal collagen from C57Bl/6 mice subjected to COPD and pre-treated with the corticosteroid dexamethasone (10 mg/kg, i.p.).

**Figure 3 nutrients-16-01509-f003:** HDAC2: Lung activity in COPD mice and protein level in human macrophages exposed to CSE. HDAC2 activity in lung tissue (**A**) of C57Bl/6 mice subjected to COPD, protein expression of HDAC2 (**B**) in human macrophages exposed to CSE, and photomicrography of HDAC2 bands (**C**) in human macrophages exposed to CSE.

**Figure 4 nutrients-16-01509-f004:** Inflammatory cells in BALF. Total and differential population of inflammatory cells in the bronchoalveolar lavage fluid (BALF) of C57Bl/6 mice subjected to COPD and pretreated with *Lacticaseibacillus rhamnosus* (Lr) (10^9^ CFU). Differential cell counting was performed by cytospin. Total cells (**A**); Macrophages (**B**); Neutrophils (**C**); Lymphocytes (**D**). Data are mean ± SEM (6–10 animals/group). ANOVA and Tukey–Kramer tests were conducted. The results were considered statistically significant when *p* < 0.05.

**Figure 5 nutrients-16-01509-f005:** Photomicrographs of lung tissue. (**A**) intrabronchial mucus, (**B**) peribronchial collagen, and (**C**) parenchymal collagen from C57Bl/6 mice subjected to COPD induction and pretreated with Lacticaseibacillus rhamnosus (Lr) (10^9^ CFU).

**Figure 6 nutrients-16-01509-f006:** Pro-inflammatory mediators in BALF. Concentration of pro-inflammatory mediators TNF-α (**A**), IL-1β (**B**), IL-6 (**C**), GM-CSF (**D**), and TSLP (**E**) in bronchoalveolar lavage fluid (BALF) and transcription factor NF-κB (**F**) in lung tissue from C57Bl/6 mice subjected to COPD and pre-treated with *Lacticaseibacillus rhamnosus* (Lr) (10^9^ CFU). BALF was collected for pro-inflammatory mediator analysis using the ELISA technique and lung tissue homogenate was collected for analysis of NF-κB using PCR. Data are mean ± SEM (6–10 animals/group). ANOVA and Tukey–Kramer tests were conducted. The results were considered statistically significant when *p* < 0.05.

**Figure 7 nutrients-16-01509-f007:** Pro-oxidant mediators in lungs. Levels of ROS (**A**), Isoprostane (**B**), NADPH oxidase (**C**) and TBARS (**D**) in the lungs of C57Bl/6 mice subjected to COPD induction and pretreated with *Lacticaseibacillus rhamnosus* (Lr) (10^9^ CFU). Lung tissue homogenate was prepared for analysis of pro-oxidant agents using enzyme. Data are mean ± SEM (6–10 animals/group). ANOVA and Tukey–Kramer tests were conducted. The results were considered statistically significant when *p* < 0.05.

**Figure 8 nutrients-16-01509-f008:** Anti-oxidant mediators in lungs. Levels of SOD (**A**), catalase (**B**), HO-1 (**C**), and transcription factor Nrf2 (**D**) in the lungs of C57Bl/6 mice subjected to COPD induction and pretreated with *Lacticaseibacillus rhamnosus* (Lr) (10^9^ CFU). Lung tissue homogenate was prepared for analysis of pro-oxidant agents using enzyme immunoassays and for analysis of Nrf2 using PCR. Data are mean ± SEM (6–10 animals/group). ANOVA and Tukey–Kramer tests were conducted. The results were considered statistically significant when *p* < 0.05.

**Figure 9 nutrients-16-01509-f009:** Cytokines and NF-κB in human macrophages. Concentration of pro-inflammatory mediators MCP-1 (**A**), MIP-1 (**B**), RANTES (**C**) and transcription factor NF-κB (**D**) in human macrophages exposed to CSE (2.5%) and pre-treated with butyrate (but) (25 μM). Supernatant of human macrophages was collected for pro-inflammatory mediator analysis using the ELISA technique and analysis of NF-κB using PCR. Data are presented as means ± SE from triplicate samples for each treatment versus control. The results were considered statistically significant when *p* < 0.05.

**Figure 10 nutrients-16-01509-f010:** Oxidative stress in human macrophages. Level of ROS (**A**), Isoprostane (**B**), TBARS (**C**), SOD (**D**), and transcription factor Nrf2 (**E**) in human macrophages exposed to CSE (2.5%) and pre-treated with butyrate (but) (25 μM). Supernatant of human macrophages was collected for pro-inflammatory mediator analysis using the ELISA technique and analysis of Nrf2 using PCR. Data are presented as means ± SE from triplicate samples for each treatment versus control. The results were considered statistically significant when *p* < 0.05.

**Figure 11 nutrients-16-01509-f011:** Influence of GPR43 receptor in in vivo effect of *Lacticaseibacillus rhamnosus*. Total cells (**A**) and macrophage population (**B**) in BALF, secretion of ROS (**C**) and NF-κB expression (**D**) in the lungs of C57Bl/6 mice subjected to COPD induction and pretreated with GLPG-094 (1 mg/kg) 1 h before the addition of *Lacticaseibacillus rhamnosus* (Lr) (10^9^ CFU). BALF was collected for analysis of both the total cells and the macrophage population. ROS analysis was performed using immunoassay technique. Lung tissue homogenate was collected for analysis of NF-κB using PCR. Data are mean ± SEM (6–10 animals/group). ANOVA and Tukey–Kramer tests were conducted. The results were considered statistically significant when *p* < 0.05.

**Figure 12 nutrients-16-01509-f012:** Influence of GPR43 receptor in in vitro effect of butyrate. Pro-inflammatory mediators MCP-1 (**A**), MIP-1 (**B**), RANTES (**C**), ROS (**D**), NF-κB (**E**), HO-1 (**F**), and Nrf2 (**G**) in human macrophages exposed to CSE (2.5%) and pretreated with GLPG-094 (0.1 mM) 1 h before the addition of butyrate (but) (25 μM). Supernatant of human macrophages was collected for pro-inflammatory mediator analysis using the ELISA technique, measurement of ROS and HO-1 using immunoassay techniques and analysis of both NF-κB and Nrf2 using PCR. Data are presented as means ± SE from triplicate samples for each treatment versus control. The results were considered statistically significant when *p* < 0.05.

**Table 1 nutrients-16-01509-t001:** Viability of human macrophages (%).

cells in culture medium:	100%
CSE:	95.33%
dexamethasone:	98.54%
butyrate:	99%
GLPG-094:	99.42%

**Table 2 nutrients-16-01509-t002:** Effect of dexamethasone on cell population in BALF of COPD mice.

Cellularity in BALF (10^4^ cells/mL)	Control	COPD	COPD + Dexamethasone
Total	5.03 ± 1.19	* 38.20 ± 1.21	(ns) 33.49 ± 4.17
Neutrophils	0.06 ± 0.05	* 25.80 ± 2.85	(ns) 21.86 ± 4.02
Macrophages	3.73 ± 0.69	* 14.72 ± 1.18	(ns) 12.18 ± 1.03
Lymphocytes	0.16 ± 0.12	* 2.26 ± 0.39	(ns) 1.97 ± 0.34

* COPD vs. control is statistically different (*p* < 0.05); COPD vs. dexamethasone is not statistically different (ns).

**Table 3 nutrients-16-01509-t003:** Effect of dexamethasone on cytokines and NF-κB in lungs of COPD mice.

Cytokines (pg/mL) andNF-κB (mRNA)	Control	COPD	COPD + Dexamethasone
TNF-α	50.15 ± 5.33	* 358.21 ± 10.25	(ns) 352.10 ± 10.12
IL-1β	20.73 ± 2.55	* 79.54 ± 7.21	(ns) 76.80 ± 7.55
IL-6	12.31 ± 2.10	* 62.18 ± 5.21	(ns) 61.57 ± 5.20
GM-CSF	55.67 + 4.22	* 252. 71 ± 5.77	(ns) 250.8 ± 6.33
TSLP	33.8 ± 1.55	* 125.81 ± 2.33	(ns) 121.92 ± 3.71
NF-κB	0.14 ± 0.01	* 1.22 ± 0.21	(ns) 1.20 ± 0.34

* COPD vs. control is statistically different (*p* < 0.05); COPD vs. dexamethasone is not statistically different (ns).

**Table 4 nutrients-16-01509-t004:** Effect of dexamethasone on airway remodeling in COPD mice.

Airway Remodeling (%)	Control	COPD	COPD + Steroid
Airway Mucus	1.15 ± 0.12	* 32.78 ± 3.63	(ns) 30.61 ± 3.60
Airway Collagen	10.22 ± 1.17	* 45.88 ± 3.20	(ns) 42.19 ± 2.77
Parenchyma Collagen	2.84 ± 0.10	* 9.64 ± 1.07	(ns) 9.55 ± 1.18

* COPD vs. control is statistically different (*p* < 0.05); COPD vs. dexamethasone is not statistically different (ns).

**Table 5 nutrients-16-01509-t005:** Effect of dexamethasone on cytokines and NF-κB in CSE-stimulated human macrophages.

Cytokines (pg/mL) NF-κB (mRNA)	Control	CSE	CSE + Dexamethasone
MCP-1	5.33 ± 0.55	* 80.94 ± 7.55	(ns) 77.50 ± 7.82
MIP-1	8.22 ± 1.18	* 36.91 ± 4.22	(ns) 35.68 ± 5.20
RANTES	7.40 ± 2.05	* 177.50 ± 8.33	(ns) 175. 30 ± 8.20
NF-κB	0.2 ± 0.01	* 0.57 ± 0.14	(ns) 0.54 ± 0.10

* COPD vs. control is statistically different (*p* < 0.05); COPD vs. dexamethasone is not statistically different (ns).

## Data Availability

The datasets generated and/or analyzed during the current study are available from the corresponding author on reasonable request.

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
