# Peer review of "Involvement of GPR43 Receptor in Effect of Lacticaseibacillus rhamnosus on Murine Steroid Resistant Chronic Obstructive Pulmonary Disease: Relevance to Pro-Inflammatory Mediators and Oxidative Stress in Human Macrophages"

_nutrients, 2024, doi:10.3390/nu16101509_

Round 1

Reviewer 1 Report

Comments and Suggestions for Authors

Smart and original article, evaluating effect of Lacticaseibacillus rhamnosus (Lr) on COPD steroid resistant model and in vivo and in vitro influence of GPR43 Receptor . The methodology is detailed precisely. Experimental method is accurately and systematically followed. Starting by description of mice induced -COPD population, introduction of Lr, the authors selected the appropriate techniques to analyse pro-inflammatory mediators, pro and anti-oxydants from the lung tissues (in vivo study). On the other hand, the same rigor was applied to the human monocytes-macrophages preparation and introduction of butyrate (in vitro study). Then the same drafts were applied on both in vivo (from mice)/in vitro (from human cells) leading to consistent results:

-          Corticosteroid Resistance (dexamethasone treatment)

-          Reduction of lung inflammation and airway remodeling by probiotic (Lr)

-          Modulation (reduction) of pro-inflammatory mediators and oxidative stress by Lr and Butyrate

-          Reduction of Lr and Butyrate effects by GPR43 inhibitor

Tables and Figures are clear and convincing, so that the discussion is structured and can support the comparison with many other publications. The bibliography is strong and well documented. The Lr and probiotic action on inflammation already described on several tissues (especially the gut) can be expanded to the lung with accurate conclusions on molecular mechanisms of steroid resistance (role of HDAC2 activity, NF-Æ™B expression…).

Some minor typo errors can be pointed:

-          Page 5, line 194: 4 mice groups listed as (1) (2) (4) (5). Should-it be (1) (2) (3) (4)?

-          Page 16, line 442: MIP-1 for Figure 12A should be MCP-1

-          Page 17, line 494: pos- transcriptional level should be post-transcriptional

-          Page 19, line 556: “ref” it seems that a reference is missing?

An error has occurred on Figures 11 and 12 description page 16, dose inversion

-          Figure 11, line 434: GLPG094 treatment is 1mg/Kg in vivo (and not 0.1mM)

-          Figure 12, line 443: GLPG094 treatment is 0.1mM in vitro (and not 1mg/Kg)

Author Response

Reviewer Comments:

 Reviewer # 1

 Page 5, line 194: 4 mice groups listed as (1) (2) (4) (5). Should-it be (1) (2) (3) (4)?

 Response: This was corrected.

 Page 16, line 442: MIP-1 for Figure 12A should be MCP-1

Response: This was corrected.

Page 17, line 494: pos- transcriptional level should be post-transcriptional

Response: This was corrected.

 Page 19, line 556: “ref” it seems that a reference is missing?

 Response: This was corrected.

 An error has occurred on Figures 11 and 12 description page 16, dose inversion

 Figure 11, line 434: GLPG094 treatment is 1mg/Kg in vivo (and not 0.1mM)

 Response: This was changed.

 Figure 12, line 443: GLPG094 treatment is 0.1mM in vitro (and not 1mg/Kg)

 Response: This was changed.

Reviewer 2 Report

Comments and Suggestions for Authors

The authors have presented a paper on GPR43 receptor in effect of Lacticaseibacillus 2 rhamnosus on murine steroid resistant COPD. The topic and results are very interesting. I have the following recommendations to improved the paper:

1. The English language of the paper needs some minor edits.

2. All tables have problems in the PDF version, please check and edit the tables.

3. Some of the refernces in the discussion section were missing.

Comments on the Quality of English Language

Minor English edits are needed.

Author Response

 Reviewer Comments:

 Reviewer # 2

 The English language of the paper needs some minor edits.

Response:  This was done.

All tables have problems in the PDF version, please check and edit the tables.

Response: This was checked and edited.

Some of the references in the discussion section were missing.

Response: This was withdrawn.
